# Binary-Integer-Programming Based Algorithm for Expert Load Balancing in Mixture-of-Experts Models

## Abstract

For pre-training of MoE (Mixture-of-Experts) models, one of the main issues is unbalanced expert loads, which may cause routing collapse or increased computational overhead. Existing methods contain the Loss-Controlled method and the Loss-Free method, where both the unbalanced degrees at first several training steps are still high and decrease slowly. In this work, we propose BIP-Based Balancing, an expert load balancing algorithm based on binary integer programming (BIP). The algorithm maintains an additional vector $q$ on each MoE layer that can help change the top-K order of $s$ by solving a binary integer programming with very small time costs. We implement the algorithm on two MoE language models: 16-expert (0.3B) and 64-expert (1.1B). The experimental results show that on both models comparing with the Loss-Controlled method and the Loss-Free method, our algorithm trains models with the lowest perplexities, while saves at least 13% of pre-training time compared with the Loss-Controlled method. Within our current knowledge, this is the first routing algorithm that achieves maintaining load balance status on every expert in every MoE layer from the first step to the last step during the whole pre-training process, while the trained MoE models also perform well.

## 1 Introduction

MoE (Mixture-of-Experts) architectures allow LLMs (Large Language Models) become sparsity so that they can have both large scale of parameters and much small resource costs. However, unbalanced expert loads always happen in MoE LLM pre-training, especially when the number of experts is large. This problem will lead to computation bottlenecks Lepikhin et al. (2021); Fedus et al. (2021) or routing collapse Shazeer et al. (2017). Furthermore, the worst situation is that a MoE model finally degenerates to a dense model, but with fewer activated parameters.

In order to balance the expert loads, many methods are proposed. One is using an auxiliary loss to encourage balanced expert load Lepikhin et al. (2021); Fedus et al. (2021). A disadvantage is that it will introduces undesired gradients that conflict with the language modeling objective, which will influence model performance. The other way is auxiliary-loss-free load balancing strategy Wang et al. (2024), where authors add an additional bias vector on routing scores to change their sort order, instead of computing auxiliary loss. Neither of the two methods can guarantee expert load balancing in the first several steps of the pre-training process, and it may cost thousands of training steps to change expert loads into balanced status Wang et al. (2024).

In this paper, we propose BIP-Based Balancing, an expert load balancing algorithm based on binary integer programming (BIP). The algorithm can be set within each routing layer, which maintains an additional vector $q$ that can help change the top-K order of $s$. The key point is that we update values of $q$ by solving a specific form of binary integer programmings on each routing gate with very small time costs. For evaluation experiments, we implement the algorithm on two MoE language models based on Minimind model series Jingyaogong (2024): 16-expert (0.3B) and 64-expert (1.1B). The experimental results show that on both models comparing with the Loss-Controlled method and the Loss-Free method, our algorithm train models with lower perplexities, while save at least 13% of pre-training time. Within our current knowledge, this is the first routing algorithm that achieves keeping load balance status on every expert and every MoE layer from the first step to the last step during the whole pre-training process, while the trained MoE models also perform well.

## 2 Preliminary

### 2.1 Mixture-of-Expert Layers in LLMs.

In MoE layers, for each token experts are selected by a Top-K routing. Let $\mathbf{u}_i$ denote the input of the $i$-th token to an $m$-expert MoE layer, then output $\mathbf{h}_i$ can be calculated as follows:

$$\mathbf{h}_i = \mathbf{u}_i + \sum_{j=1}^{m} g_{ij} \, \text{FFN}_j \left( \mathbf{u}_i \right),$$

$$g_{ij} = \begin{cases} s_{ij}, & s_{ij} \in \text{Topk} \left( \{ s_{it} \mid 1 \le t \le m \}, k \right), \\ 0, & \text{otherwise}, \end{cases}$$

$$s_{ij} = G \left( \mathbf{u}_i^T \mathbf{e}_j \right).$$

Here $G$ is a nonlinear gating function and $\mathbf{e}_j$ is the centroid of the $j$-th expert.

### 2.2 Load Balancing Strategy with Auxiliary Loss (Loss-Controlled Method)

Auxiliary-loss methods have been used for control load balance Lepikhin et al. (2021); Fedus et al. (2021). For a sequence with length $n$, its auxiliary loss is calculated by

$$\mathcal{L}_{\text{Balance}} = \alpha \sum_{j=1}^{m} f_j P_j,$$

$$f_j = \frac{m}{kn} \sum_{i=1}^{n} \delta_{ij},$$

$$P_j = \frac{1}{n} \sum_{i=1}^{n} s_{ij},$$

where $m$ is the total number of experts, $k$ is the number of experts selected for each token, $s_{ij}$ is the routing score of Expert $j$ for Token $i$, $f_j$ represents the fraction of tokens routed to Expert $j$. $\delta_{ij}$ is 0 or 1 representing whether Token $i$ is for Expert $j$. $P_j$ denotes the average gating scores of Expert $j$, and $\alpha$ is a hyper-parameter controlling the strength of the auxiliary loss.

### 2.3 Auxiliary-Loss-Free Load Balancing Strategy

The other way to expert load balance is auxiliary-loss-free load balancing strategy Wang et al. (2024), which first appears in DeepSeek-V3 DeepSeek-AI et al. (2024). Instead of computing loss functions, the authors introduce a bias vector $\boldsymbol{b}$ on expert lists so that it can influence the determination of the top-K routing as follows:

$$g'_{ij} = \begin{cases} s_{ij}, & s_{ij} + b_j \in \text{Topk}(\{ s_{it} + b_t \mid 1 \le t \le m \}, k), \\ 0, & \text{otherwise}. \end{cases}$$

## 3 Algorithm

The main result of this work is BIP-Based Balancing algorithm, whose details are described in Algorithm 1. Like Loss-Free strategy, BIP-Based Balancing algorithm also does not need to compute auxiliary-loss, while there is also an additional vector $\boldsymbol{q}$ that can help change the top-K order of $\boldsymbol{s}$. The main difference is that the value of $\boldsymbol{q}$ is computed by solving a binary integer programming, and we update values of $\boldsymbol{q}$ after each calculation of a routing gate, instead of after each batch. Without loss of generality, all vectors mentioned in algorithms are row vectors.

To explain why Algorithm 1 works, we first model the expert load balancing problem to the following binary integer programming (BIP) problem:

---

**Algorithm 1** BIP-Based Balancing Algorithm for MoE Models

---

1: **Input:** MoE model $\boldsymbol{\theta}$, expert number $m$, topk number $k$ and a small constant $T$.
2: Initialize $q_{lj} = 0$ for each expert $j$ in every MoE layer $l$;
3: **for** a batch $\{\boldsymbol{x}, \boldsymbol{y}\}$ in the batch enumeration **do**
4:     Set $n$ be the token number of $\{\boldsymbol{x}, \boldsymbol{y}\}$;
5:     **for** each MoE layer $l$ **do**
6:         Compute the routing score matrix $\boldsymbol{s} \in \mathbb{R}^{n \times m}$ on the batch data $\{\boldsymbol{x}, \boldsymbol{y}\}$ and all experts;
7:         **for** $t = 1, ..., T$ **do**
8:             Set $\boldsymbol{P} = \boldsymbol{s} - \boldsymbol{1}_n^T \boldsymbol{q}_l \in \mathbb{R}^{n \times m}$;
9:             Set $p_i = \max(0, (k+1)$ −th largest value of $\boldsymbol{P}_i), 1 \le i \le n$ for $\boldsymbol{p} \in \mathbb{R}^n$;
10:            Set $\boldsymbol{Q} = \boldsymbol{s}^T - \boldsymbol{1}_m^T \boldsymbol{p} \in \mathbb{R}^{m \times n}$;
11:            Set $q_{lj} = \max(0, (nk/m + 1)$ −th largest value of $\boldsymbol{Q}_j), 1 \le j \le m$;
12:        **end for**
13:        For every token $i$ and expert $j$, set

$$g_{ij} = \begin{cases} s_{ij}, & s_{ij} - q_{lj} \in \text{Topk}(\{s_{it} - q_{lt} | 1 \le t \le m\}, k), \\ 0, & \text{otherwise.} \end{cases}$$

14:        Continue pre-training process on $\boldsymbol{\theta}$ with the expert decision matrix $\boldsymbol{g}$;
15:    **end for**
16: **end for**
17: **Output:** trained model $\boldsymbol{\theta}$.

---

$$\max \quad \sum_{i=1}^{n} \sum_{j=1}^{m} s_{ij} x_{ij} \tag{BIP}$$

$$\text{s.t.} \quad \sum_{j=1}^{m} x_{ij} \le k, \forall i \in [n] \tag{1}$$

$$\sum_{i=1}^{n} x_{ij} \le \frac{kn}{m}, \forall j \in [m] \tag{2}$$

$$x_{ij} \in \{0, 1\}, \forall i \in [n], \forall j \in [m].$$

Here, $n$ is the number of tokens in one batch, $m$ is the number of experts and $k$ is the number of experts selected by each routing decision. $\boldsymbol{s}$ is the routing score matrix. The binary decision variables $x_{ij}$ determine whether matching token $i$ with expert $j$. Condition (1) holds since we can only match one token with $k$ experts. Condition (2) ensures the load balance of experts.

in order to solve (BIP), consider its linear programming relaxation version:

$$\max \quad \sum_{i=1}^{n} \sum_{j=1}^{m} s_{ij} x_{ij} \tag{P-LP}$$

$$\text{s.t.} \quad \sum_{j=1}^{m} x_{ij} \le k, \forall i \in [n]$$

$$\sum_{i=1}^{n} x_{ij} \le \frac{kn}{m}, \forall j \in [m]$$

$$\boldsymbol{0} \le \boldsymbol{x} \le \boldsymbol{1}.$$

The dual problem of (P-LP) is:

$$\min \quad k \sum_{i=1}^{n} p_i + \frac{kn}{m} \sum_{j=1}^{m} q_j + \sum_{i=1}^{n} \sum_{j=1}^{m} r_{ij} \qquad \text{(D-LP)}$$
$$\text{s.t.} \quad p_i + q_j + r_{ij} \geq s_{ij}, \forall i \in [n], \forall j \in [m]$$
$$\boldsymbol{p} \geq \boldsymbol{0}, \boldsymbol{q} \geq \boldsymbol{0}, \boldsymbol{r} \geq \boldsymbol{0},$$

where the decision variables are $\boldsymbol{p} \in \mathbb{R}^n$, $\boldsymbol{q} \in \mathbb{R}^m$ and $\boldsymbol{r} \in \mathbb{R}^{n \times m}$.

By primal-dual principle, we have the inequality (BIP)≤(P-LP)=(D-LP). In fact, the optimal solution $\boldsymbol{x^*}$ of (P-LP) has the following relationship between the optimal solution $\boldsymbol{p}^*, \boldsymbol{q}^*$ of (D-LP):

$$x_{ij}^* = 1 \text{ if and only if } p_i^* + q_j^* < s_{ij}.$$

This result matches the line 13 in Algorithm 1, since we can change the inequality $p_i^* + q_j^* < s_{ij}$ to the form $s_{ij} - q_j^* > p_i^*$. Thus when $i$ is fixed, there are exact $m - k$ subscripts $j$ satisfying $s_{ij} - q_j^* <= p_i^*$ while the other $k$ subscripts $j$ satisfy $s_{ij} - q_j^* > p_i^*$, which exactly match the subscripts of Topk($\{s_{ij}^* - q_j^* | 1 \leq j \leq m\}, k$).

On solving (D-LP), we use the standard ADMM algorithm Boyd et al. (2010):

---

**Algorithm 2** ADMM algorithm for (D-LP)

---

1: **for** $t = 1, ..., T$ **do**
2:      Set $\boldsymbol{p}_t = \arg\max_{\boldsymbol{p}} L_\lambda(\boldsymbol{p}, \boldsymbol{q}_{t-1}, \boldsymbol{r}_{t-1}, \boldsymbol{u}_{t-1})$;
3:      Set $\boldsymbol{q}_t = \arg\max_{\boldsymbol{q}} L_\lambda(\boldsymbol{p}_t, \boldsymbol{q}, \boldsymbol{r}_{t-1}, \boldsymbol{u}_{t-1})$;
4:      Set $\boldsymbol{r}_t = \arg\max_{\boldsymbol{r}} L_\lambda(\boldsymbol{p}_t, \boldsymbol{q}_t, \boldsymbol{r}, \boldsymbol{u}_{t-1})$;
5:      Update $\boldsymbol{u}_t$ with the step parameter $\lambda$;
6: **end for**

---

Here $L_\lambda(\boldsymbol{p}, \boldsymbol{q}, \boldsymbol{r}, \boldsymbol{u})$ is the augmented Lagrangian function of (D-LP) and $\boldsymbol{u}$ is the dual vector variable in $L$. In order to implement optimizations, first notice that when $\boldsymbol{p}, \boldsymbol{q}$ are fixed, the optimal values of $\boldsymbol{r}$ and $\boldsymbol{u}$ are $r_{ij}^* = \max(s_{ij} - p_i - q_j, 0)$ and $\boldsymbol{u}^* = \boldsymbol{0}$. Then it is easy to verify that the line 2 and line 3 in Algorithm 2 imply the line 7 to line 12 part in Algorithm 1, by noticing that when $\boldsymbol{q}$ and $i$ are fixed, in order to keep exact $k$ of $\{x_{ij}\}_{1 \leq j \leq m}$ satisfying $x_{ij} > 0$, we only need to keep exact $k$ of inequalities $\{p_i + q_j < s_{ij}\}_{1 \leq j \leq m}$ hold. That is, the best choice of $p_i$ is the $(k+1)$-th largest value of $\{s_{ij} - q_j\}_{1 \leq j \leq m}$. The analysis of optimizing $\boldsymbol{q}$ when $\boldsymbol{p}$ is fixed is similar.

## 4 Experiments

### 4.1 Experimental Settings

Model Architectures, Training Settings and Hardware. The models we choose in the experiments are from Minimind, a popular extremely-lightweight language model series Jingyaogong (2024). We train 2 models on its MoE version during the experiments, one is with 16 experts and the other is with 64 experts. For the MoE version of Minimind, the number of parameters in each expert is less than 20M, and the core function of MoE gates are $softmax$. The datasets are also from Jingyaogong (2024), where we split the pre-training dataset into a training set and a test set. In order to compare time cost efficiency between different routing algorithms more intuitively, we do not shuffle the datasets (See also the mutations of blue and green lines in Figure 1 and Figure 2). More information of models, settings and GPUs is listed in Table 1.

| Hyper parameters | 16-expert model | 64-expert model |
|---|---|---|
| Vocab size | 6400 | 6400 |
| Max Sequence Length | 8192 | 8192 |
| Attention heads | 8 | 8 |
| routing score function | $softmax$ | $softmax$ |
| MoE layers | 8 | 8 |
| Routed experts | 4 | 8 |
| Activated routed experts | 16 | 64 |
| Model size | 0.3B | 1.1B |
| GPUs | RTX4090 $\times 1$ | L20 $\times 1$ |

Table 1: Model architectures, training settings and hardware.

**Baseline.** We compare our BIP-Balancing algorithm with both Loss-Controlled and Loss-Free strategies, especially in Loss-Controlled method since there are discussions show that on the $softmax$ function the Loss-Controlled method works better Su (2025). For the baseline, we set $\alpha = 0.1$ for the Loss-Controlled method which is the same value in the original Minimind model, and set $u = 0.001$ for the Loss-Free method which is supported in Wang et al. (2024).

**Measurements.** We introduce two measurements, Average Maximum Violation ($AvgMaxVio$) and Supremum Maximum Violation ($SupMaxVio$), to measure the balance degree of experts among the whole pre-training process. $AvgMaxVio$ is the average value of $MaxVio_{batch_i}$ among all training batches:

$$AvgMaxVio = \frac{\sum_{batch_i \in Batches} MaxVio_{batch_i}}{|Batches|},$$

and $SupMaxVio$ is the maximum value of $MaxVio_{batch_i}$ among all training batches:

$$SupMaxVio = \max_{batch_i \in Batches} \{MaxVio_{batch_i}\}.$$

Here

$$MaxVio_{batch_i} = \frac{\max_j Load_{ij}}{\overline{Load}} - 1,$$

where $Load_{ij}$ represents the number of tokens matched to the $j$-th expert during the $i$-th batch pre-training, and $\overline{Load}$ denotes the average number of tokens per expert in one batch. ($MaxVio_{batch}$ is first introduced in Wang et al. (2024).) The less $AvgMaxVio$ is, the faster expert loads turn into balance states, which will lead to smaller time costs of LLM training and higher computing resource utilization. On the other hand, when $SupMaxVio$ is small (for example, less than 0.2), then global training process can be seen as a balanced status approximately. Moreover, we will also show $AvgMaxVio$ of each layer in the models in Appendix A.

Besides, we also use Perplexity to measure the performances of pre-trained models, and Training time to measure the efficiency of global training processes.

### 4.2 Main Results

Table 2 shows that on the 16-expert model, comparing with the Loss-Controlled method, the BIP-based algorithm with 4 different iteration times all achieve lower perplexities. More precisely, the BIP-based algorithm with $T = 4$ (which has lowest perplexity) only cost 86.83% training time of which the Loss-Controlled method costs. This is due to the much lower values of $AvgMaxVio$ and $SupMaxVio$ (0.0602 versus 0.3852, and 0.1726 versus 1.5245). More details on the whole pre-training process are shown in Figure 1, where the $MaxVio_{batch_i}$ of Loss-Controlled method process has a large fluctuation, while the BIP-based method help maintain a smooth state on $MaxVio_{batch_i}$ of the whole pre-training process.

| Algorithm | $AvgMaxVio$ | $SupMaxVio$ | Perplexity | Training time/h |
|---|---|---|---|---|
| Loss-Controlled | 0.3852 | 1.5245 | 12.4631 | 4.6126 |
| Loss-Free | 0.1275 | 1.7702 | 11.1311 | 4.3558 |
| BIP, $T = 2$ | 0.0529 | 0.2019 | 11.2417 | 3.9547 |
| BIP, $T = 4$ | 0.0602 | 0.1726 | 10.6856 | 4.0051 |
| BIP, $T = 8$ | 0.0626 | 0.1727 | 10.7291 | 4.0623 |
| BIP, $T = 14$ | 0.0547 | 0.1925 | 10.7408 | 4.177 |

Table 2: Evaluation results on training MoE models with $m = 16$ and $k = 4$.

Table 3 shows that on the 64-expert model, comparing with the Loss-Controlled method, the BIP-based algorithm with $T = 14$ achieves lower perplexities, while perplexities of other 3 BIP-based algorithms with different iteration times are almost the same. The BIP-based algorithm with $T = 14$ only cost 86.15% training time of which the Loss-Controlled method costs. It's important to emphasize that, unlike Loss-Controlled and Loss-Free methods, the $AvgMaxVio$ and $SupMaxVio$ of BIP-Based algorithm do not increase fast from the 16-expert model to the 64-expert one, which still remain at a low level. More details on the whole pre-training process are shown in Figure 2. Notice that the separations among three colors of lines are more obvious than the ones in Figure 1.

| Algorithm | $AvgMaxVio$ | $SupMaxVio$ | Perplexity | Training time/h |
|---|---|---|---|---|
| Loss-Controlled | 0.7158 | 2.3841 | 9.9956 | 23.7726 |
| Loss-Free | 0.3366 | 2.7121 | 10.2975 | 23.9557 |
| BIP, $T = 2$ | 0.0513 | 0.5613 | 10.6916 | 20.4569 |
| BIP, $T = 4$ | 0.0496 | 0.4107 | 10.1299 | 20.3046 |
| BIP, $T = 8$ | 0.0441 | 0.2372 | 10.0677 | 20.4572 |
| BIP, $T = 14$ | 0.0529 | 0.1946 | 9.9071 | 20.4799 |

Table 3: Evaluation results on training MoE models with $m = 64$ and $k = 8$.

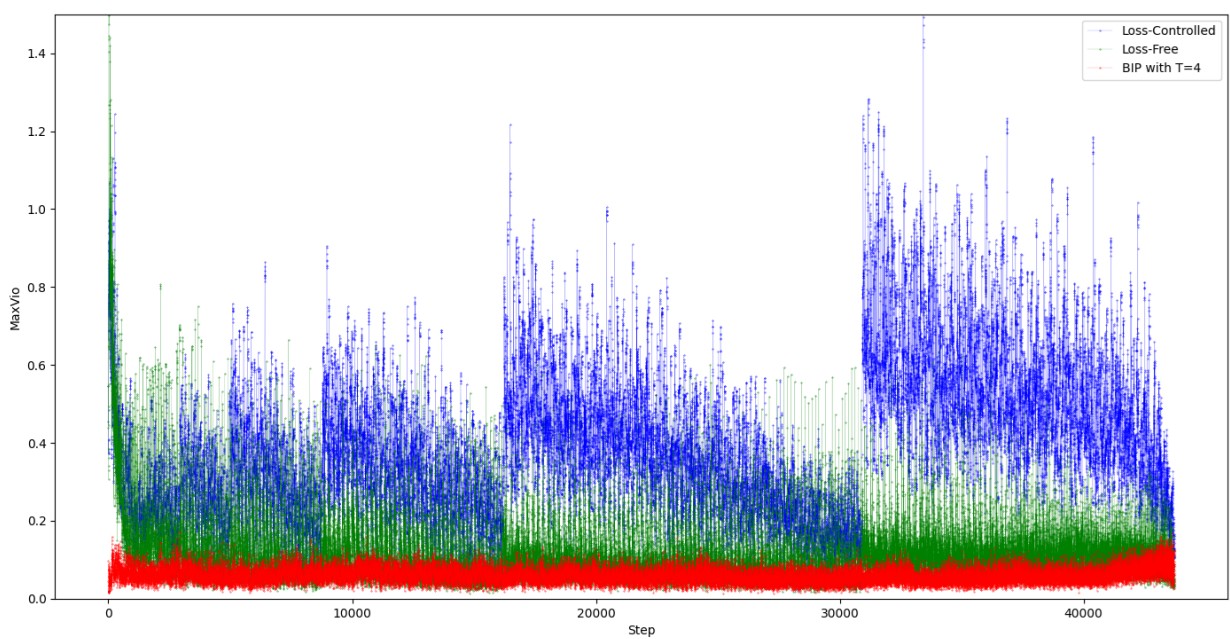

Figure 1: The line graph of relationships between training steps and $MaxVio_{batch_i}$ by different methods in the 16-expert model. The blue lines and dots represent trends of $MaxVio_{batch_i}$ by the Loss-Controlled method. The green lines and dots represent trends of $MaxVio_{batch_i}$ by the Loss-Free method. The red lines and dots represent trends of $MaxVio_{batch_i}$ by the BIP-based method.

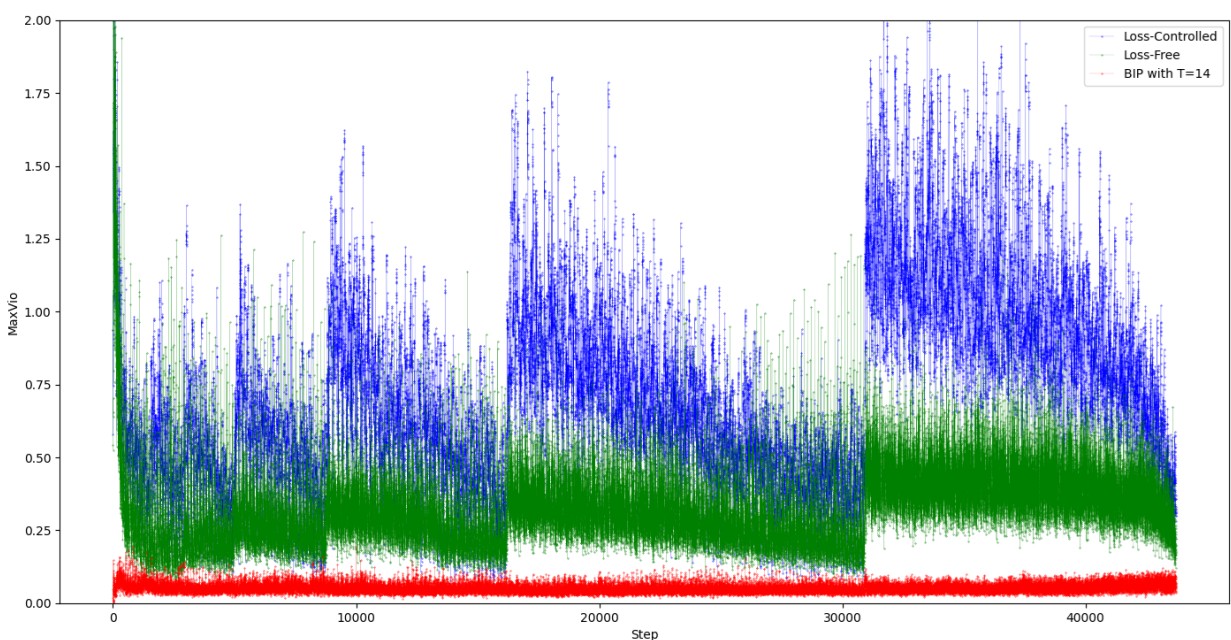

Figure 2: The line graph of relationships between training steps and $MaxVio_{batch_i}$ by different methods in the 64-expert model. The blue lines and dots represent trends of $MaxVio_{batch_i}$ by the Loss-Controlled method. The green lines and dots represent trends of $MaxVio_{batch_i}$ by the Loss-Free method. The red lines and dots represent trends of $MaxVio_{batch_i}$ by the BIP-based method.

The similar conclusions also hold for each layer in both two models. For more information on experimental data and statistical charts, see Appendix A.

## 5 Discussion

### 5.1 Online Algorithm for Problem (BIP)

We can easily provide Algorithm 3, the online version of Algorithm 1 on one routing gate:

---

**Algorithm 3** BIP-Based Balancing Algorithm for MoE Models (Online Version, on one gate)

---

1: Input: token number per batch $n$, expert number $m$, topk number $k$ and a small constant $T$
2: Initialize $\boldsymbol{q} = \boldsymbol{0}^m$, $\boldsymbol{Q} = \{Q_j = \phi, 1 \leq j \leq m\}$
3: for a token arrives at this routing gate do
4:     Get routing scores $\{s_1, ..., s_m\}$
5:     for $j = 1, ..., m$ do
6:         Set
$$g_j = \begin{cases} s_j, & s_j - q_j \in \text{Topk}(\{s_l - q_l | 1 \leq l \leq m\}, k), \\ 0, & \text{otherwise.} \end{cases}$$
7:     end for
8:     for $t = 1, ..., T$ do
9:         Set $p = \max(0, (k+1) -$th largest value of $\{s_l - q_l | 1 \leq l \leq m\})$
10:         for $j = 1, ..., m$ do
11:             Set $q_j = \max(0, (nk/m + 1) -$th largest value of $\{Q_j \cup \{s_j - p\}\})$
12:         end for
13:     end for
14:     Set $\boldsymbol{Q} = \{Q_j \cup \{s_j - p\}, 1 \leq j \leq m\}$
15: end for

---

Algorithm 3 can be applied in recommendation systems. Consider a scenario that there are over one advertisement slots on one webpage. If our aim is to maximize the sum of CTRs and restrict flows of the most popular advertisement provider, then the problem turns to be (BIP). (In this problem, an expert can be seen as a slot.) Furthermore, the approximation version of Algorithm 3 is a better choice for this model (see Algorithm 4 in Section 5.2), since its space complexity has no relationship with the number of flows.

In fact, we notice that this scenario is a special case of multi-slots online matchings Lu et al. (2022) . The online matching problem has been widely studied, but for its multi-slot version, there is only a few works Lu et al. (2022). The difficulty is that it is non-trivial to extend existing methods to the multi-slots version with diversity pursuit Zhang (2009); Yan et al. (2020). They either depend on closed-form computations Zhong et al. (2015); Agrawal et al. (2018) or additional assumptions which fail to hold in more than one slots Lu et al. (2021); Balseiro et al. (2021). We believe that our algorithms in this work will help on solving this difficult problem.

## 5.2 Approximate Algorithm with Constant Space Complexity

For Algorithm 3 there are some issues with time and space complexities need to be discussed, especially on maintaining the set array $\boldsymbol{Q}$. For each $Q_j \in \boldsymbol{Q}$, we can use a heap to maintaining its $(nk/m)$-largest member. Thus for each token, the time complexity of maintaining $\boldsymbol{Q}$ and $\boldsymbol{q}$ is only $\mathrm{O}(m \log n)$, or $\mathrm{O}(\log n)$ per expert on parallel computing. However, we will need $\mathrm{O}(m * (nk/m)) = \mathrm{O}(nk)$ space to storage sets in $\boldsymbol{Q}$, which can be seen as a linear relationship with the token size (or the number of flows). Since in recommendation situations, the scale of flows per day can be over millions, which will cost too much storage resources on running Algorithm 3.

In order to fix this issue, we notice that if $\boldsymbol{0 < s < 1}$ holds, we can divide the internal $[0, 1)$ into several blocks. Instead of maintaining the set array, we only need to count numbers lying in each block. when we update the vector $\boldsymbol{q}$, we first find the block that $(nk/m + 1)$-th largest number lies in, then use interpolation to approximate its value. Algorithm 4 shows details. Notice that the space complexity of Algorithm 4 is $\mathrm{O}(m)$, which has no relationship with the token number.

---

**Algorithm 4** BIP-Based Balancing Algorithm for MoE Models (Online Approximation Version)

---

1: Input: token number $n$, expert number $m$, topk number $k$, constant $b$ and $T$
2: Initialize $\boldsymbol{q} = \boldsymbol{0}^m$, $\boldsymbol{Q} = \boldsymbol{0}^{mb}$
3: for a token arrives at this routing gate do
4:     Get routing scores $\{s_1, ..., s_m\}$
5:     for $j = 1, ..., m$ do
6:         Set
$$g_j = \begin{cases} s_j, & s_j - q_j \in \mathrm{Topk}(\{s_l - q_l | 1 \le l \le m\}, k), \\ 0, & \text{otherwise.} \end{cases}$$
7:     end for
8:     for $t = 1, ..., T$ do
9:         Set $p = \max(0, (k+1) -\text{th largest value of } \{s_l - q_l | 1 \le l \le m\})$
10:         for $j = 1, ..., m$ do
11:             Set
$$Q'_{jl} = \begin{cases} Q_{jl} + 1, & s_j - p \ge 0 \text{ and } \frac{l}{b} \le s_j - p < \frac{l+1}{b}, \\ Q_{jl}, & \text{otherwise.} \end{cases}$$
12:             Set
$$q_j = \begin{cases} \text{interpolation between } \frac{l}{b} \text{ and} \frac{l+1}{b}, & \exists l, (nk/m + 1)-\text{th largest of } Q'_j \text{ is in } [\frac{l}{b}, \frac{l+1}{b}), \\ 0, & \text{otherwise.} \end{cases}$$
13:         end for
14:     end for
15:     Set $\boldsymbol{Q} = \boldsymbol{Q'}$
16: end for

---

## 6 Conclusion

In this work we provide BIP-Based Balancing, an expert load balancing algorithm based on binary integer programming (BIP). The algorithm keep expert load balance by solving a specific form of binary integer programmings with small time costs. The experimental results show BIP-based algorithm achieves keeping load balance status on every expert and every MoE layer from the first step to the last step during the whole pre-training process, while the trained MoE models also perform well. Finally, we discuss the potential applications of BIP-based algorithm in the fields of recommendation system and online matching.

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

## A   Other Tables and Graphics in Section 4.2

| Algorithm | Layer 1 | Layer 2 | Layer 3 | Layer 4 | Layer 5 | Layer 6 | Layer 7 | Layer 8 |
|---|---|---|---|---|---|---|---|---|
| Auxiliary Loss | 0.8988 | 1.1607 | 1.1717 | 1.1726 | 1.1528 | 1.14 | 1.1403 | 1.1216 |
| Loss Free | 0.364 | 0.3044 | 0.3341 | 0.3556 | 0.3279 | 0.4681 | 0.4827 | 0.3693 |
| BIP, $T = 4$ | 0.2024 | 0.1314 | 0.1722 | 0.2153 | 0.1584 | 0.1879 | 0.1998 | 0.2065 |

Table 4: $AvgMaxVio$ on each layer in MoE models with $m = 16$ and $k = 4$ achieved by different routing algorithms, for expert load balance evaluations.

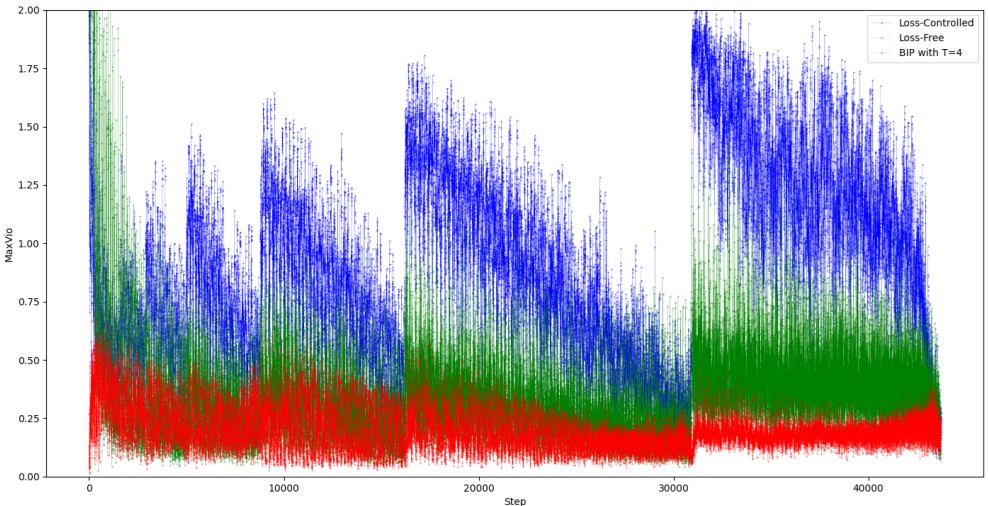

Figure 3: The line graph of relationships between training steps and $MaxVio_{batch_i}$ by different methods in the 16-expert model on layer 1.

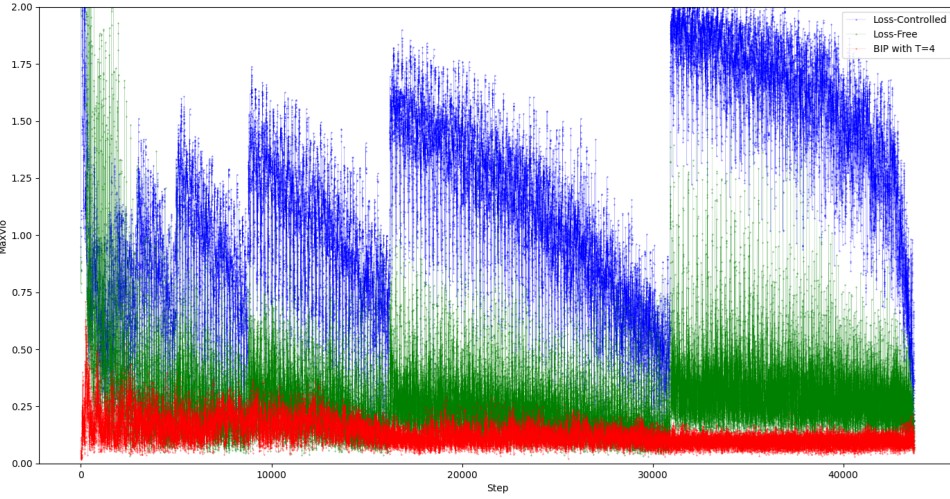

Figure 4: The line graph of relationships between training steps and $MaxVio_{batch_i}$ by different methods in the 16-expert model on layer 2.

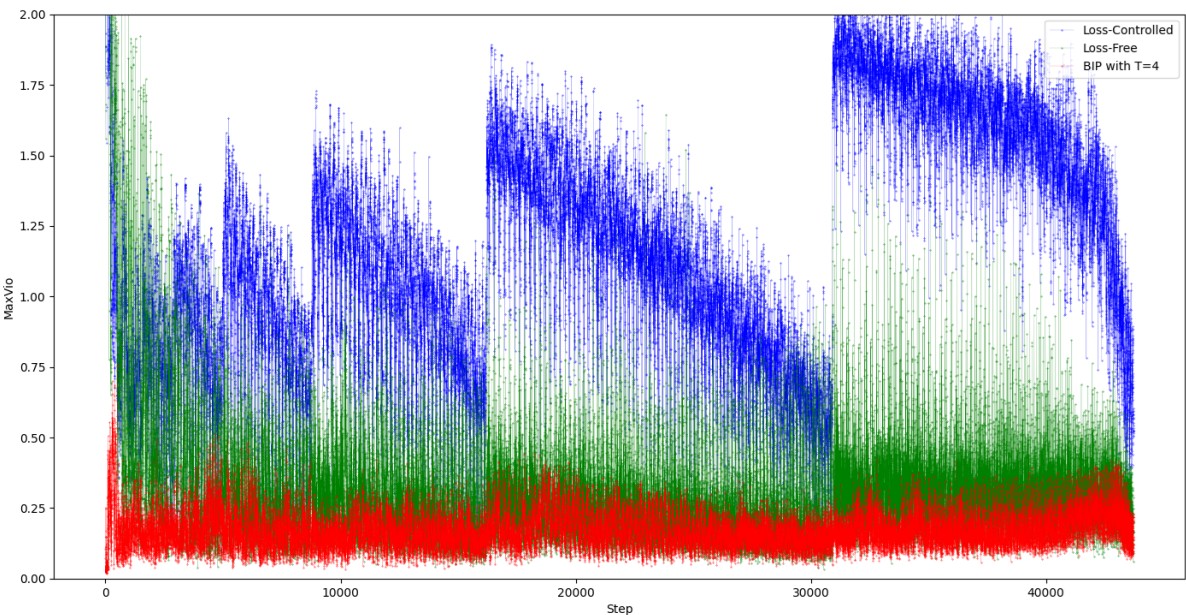

Figure 5: The line graph of relationships between training steps and $MaxVio_{batch_i}$ by different methods in the 16-expert model on layer 3.

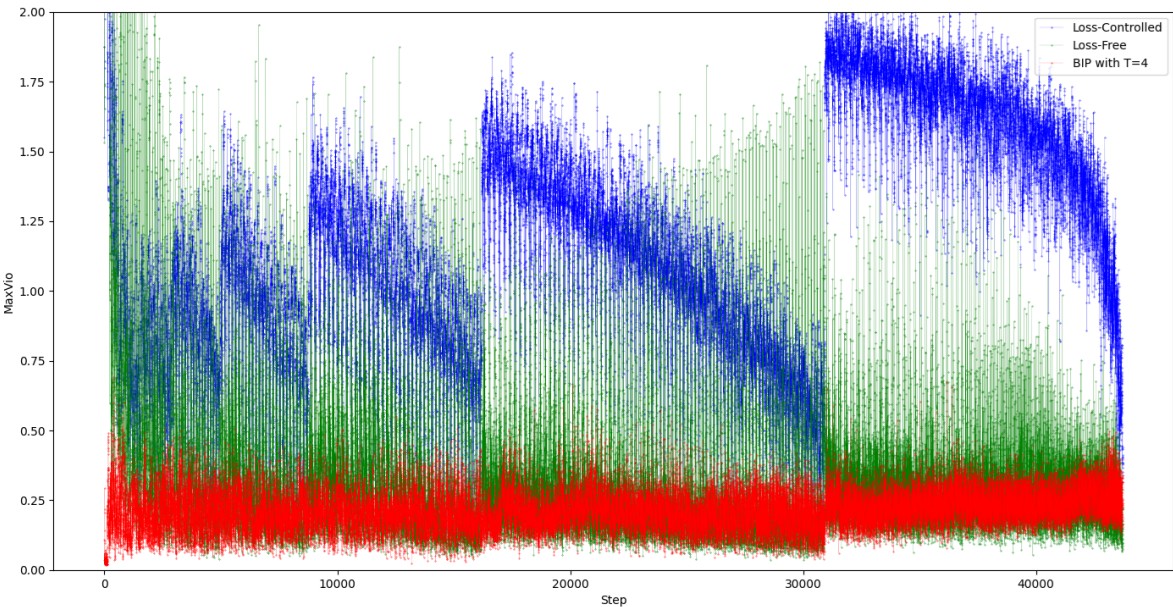

Figure 6: The line graph of relationships between training steps and $MaxVio_{batch_i}$ by different methods in the 16-expert model on layer 4.

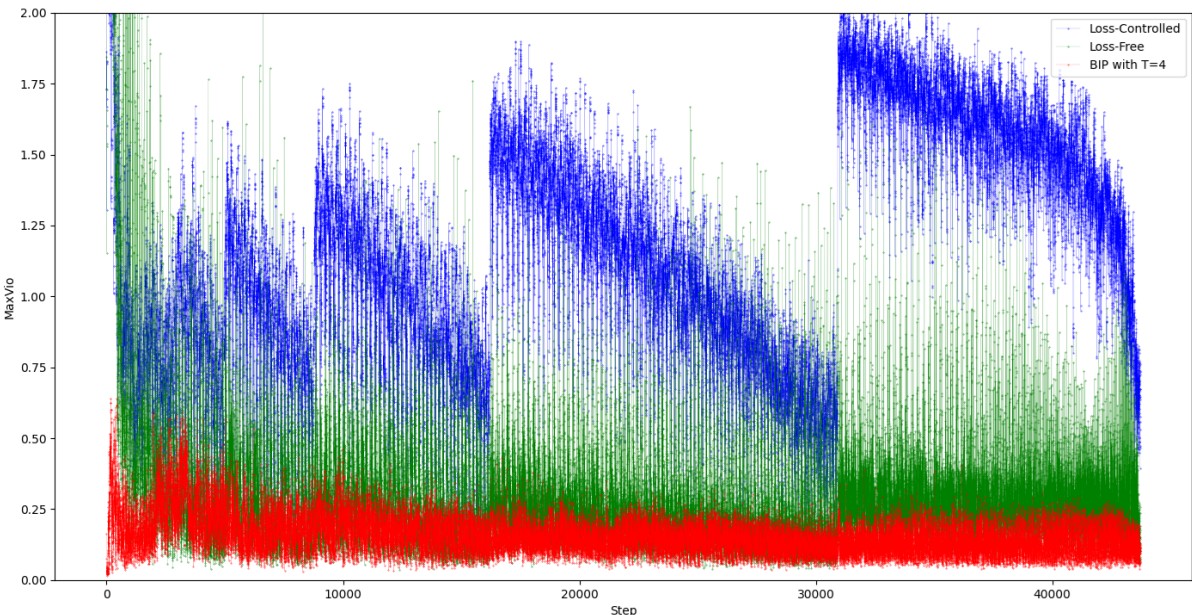

Figure 7: The line graph of relationships between training steps and $MaxVio_{batch_i}$ by different methods in the 16-expert model on layer 5.

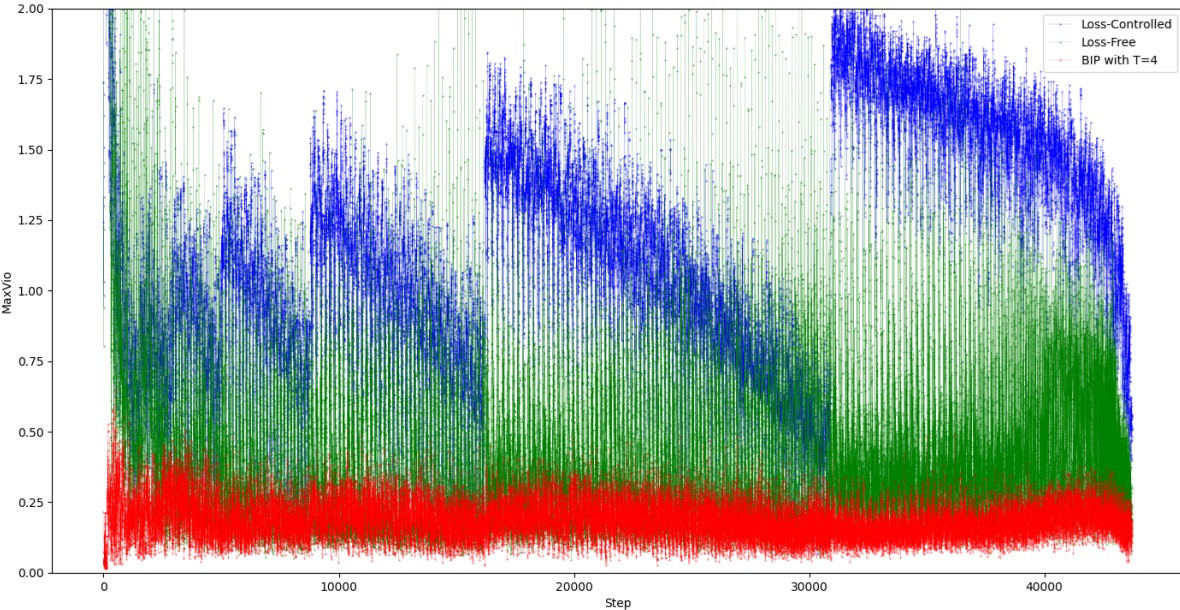

Figure 8: The line graph of relationships between training steps and $MaxVio_{batch_i}$ by different methods in the 16-expert model on layer 6.

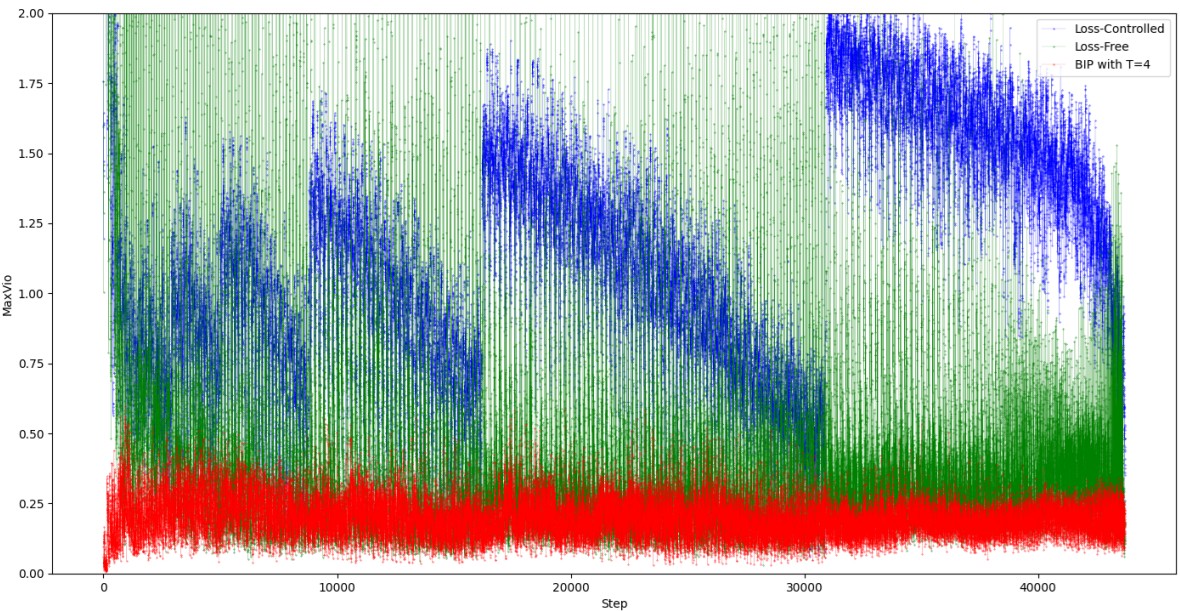

Figure 9: The line graph of relationships between training steps and $MaxVio_{batch_i}$ by different methods in the 16-expert model on layer 7.

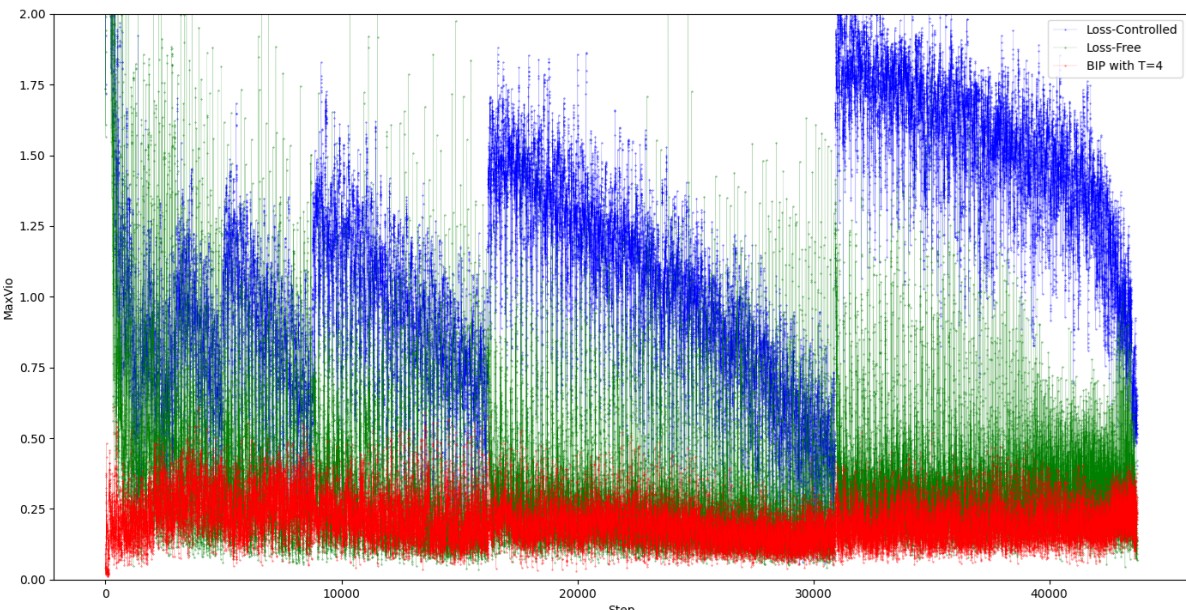

Figure 10: The line graph of relationships between training steps and $MaxVio_{batch_i}$ by different methods in the 16-expert model on layer 8.

| Algorithm | Layer 1 | Layer 2 | Layer 3 | Layer 4 | Layer 5 | Layer 6 | Layer 7 | Layer 8 |
|---|---|---|---|---|---|---|---|---|
| Auxiliary Loss | 2.469 | 2.4456 | 2.4983 | 2.478 | 2.4586 | 2.3725 | 2.2958 | 2.177 |
| Loss Free | 1.5253 | 1.0639 | 1.0399 | 1.0587 | 1.036 | 1.1521 | 1.1314 | 1.1126 |
| BIP, $T = 14$ | 0.1676 | 0.1138 | 0.1133 | 0.1109 | 0.1342 | 0.1356 | 0.2743 | 0.1888 |

Table 5: $AvgMaxVio$ on each layer in MoE models with $m = 64$ and $k = 8$ achieved by different routing algorithms, for expert load balance evaluations.

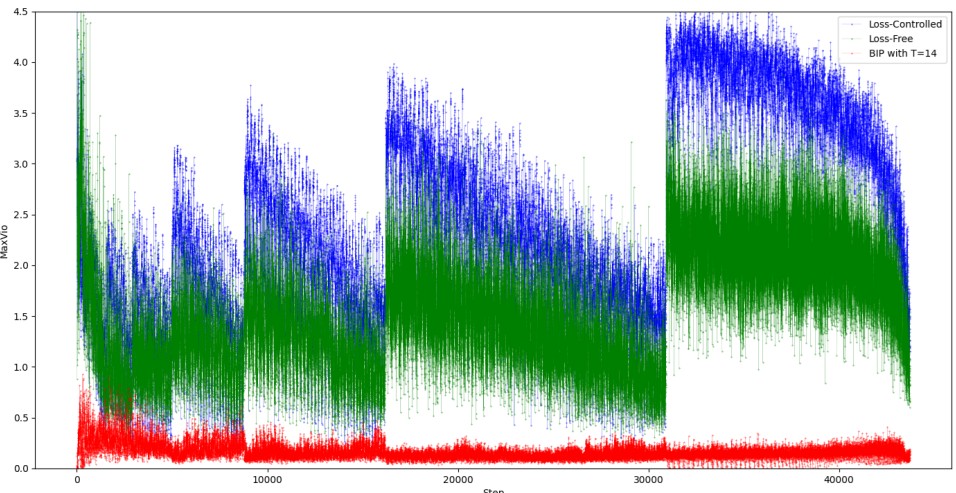

Figure 11: The line graph of relationships between training steps and $MaxVio_{batch_i}$ by different methods in the 64-expert model on layer 1.

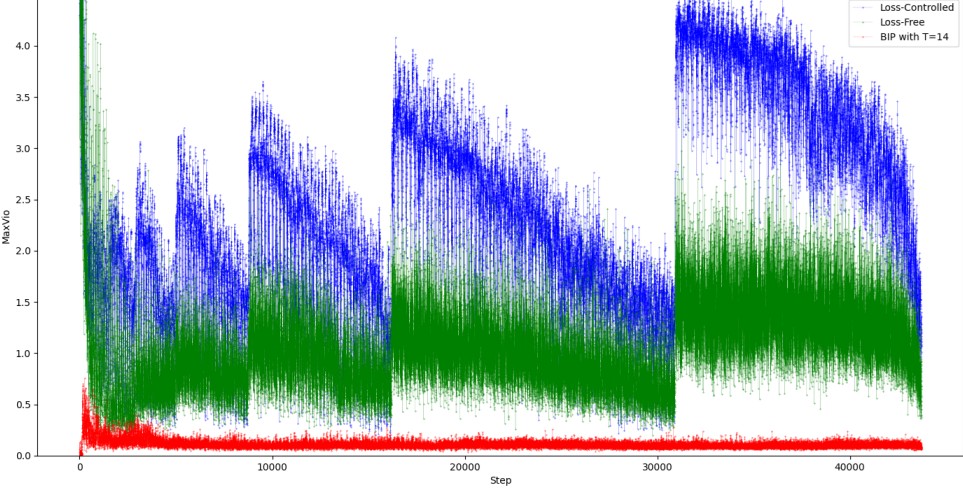

Figure 12: The line graph of relationships between training steps and $MaxVio_{batch_i}$ by different methods in the 64-expert model on layer 2.

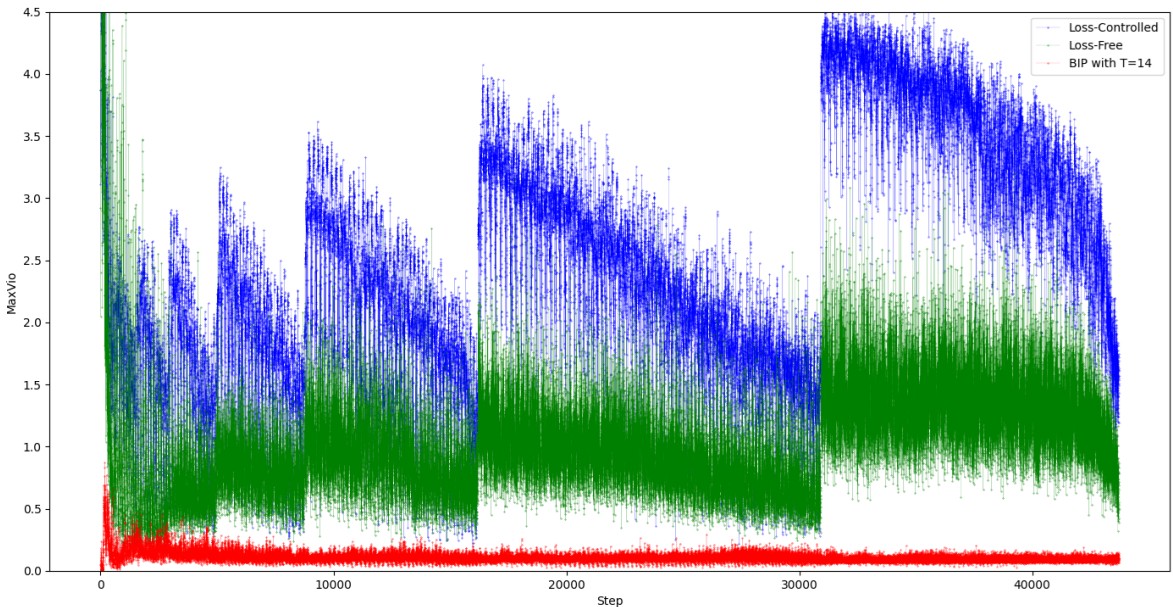

Figure 13: The line graph of relationships between training steps and $MaxVio_{batch_i}$ by different methods in the 64-expert model on layer 3.

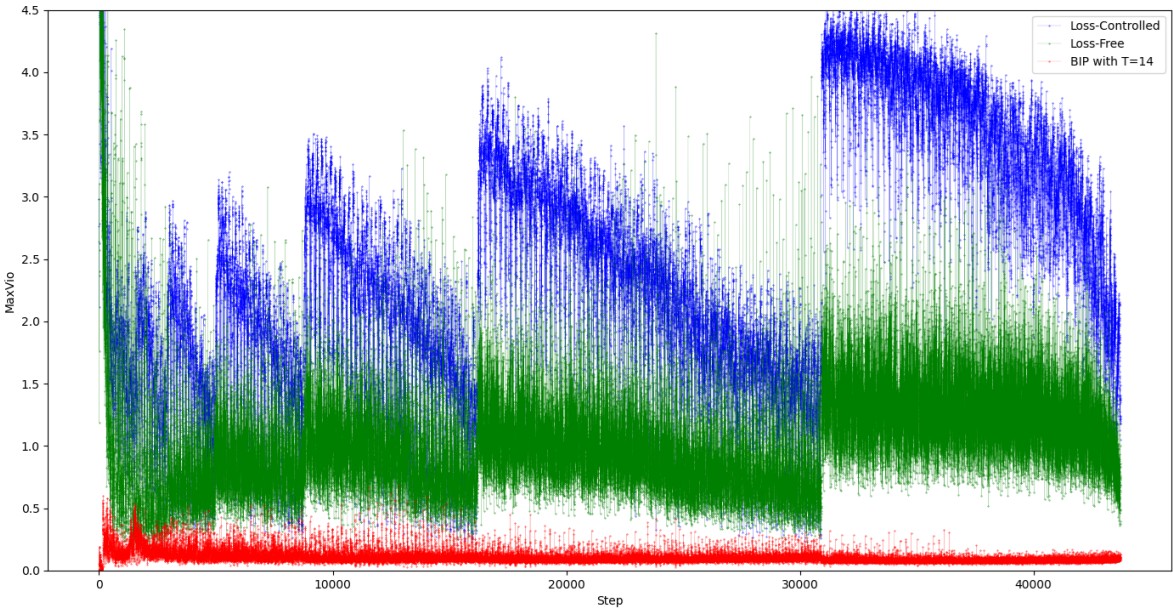

Figure 14: The line graph of relationships between training steps and $MaxVio_{batch_i}$ by different methods in the 64-expert model on layer 4.

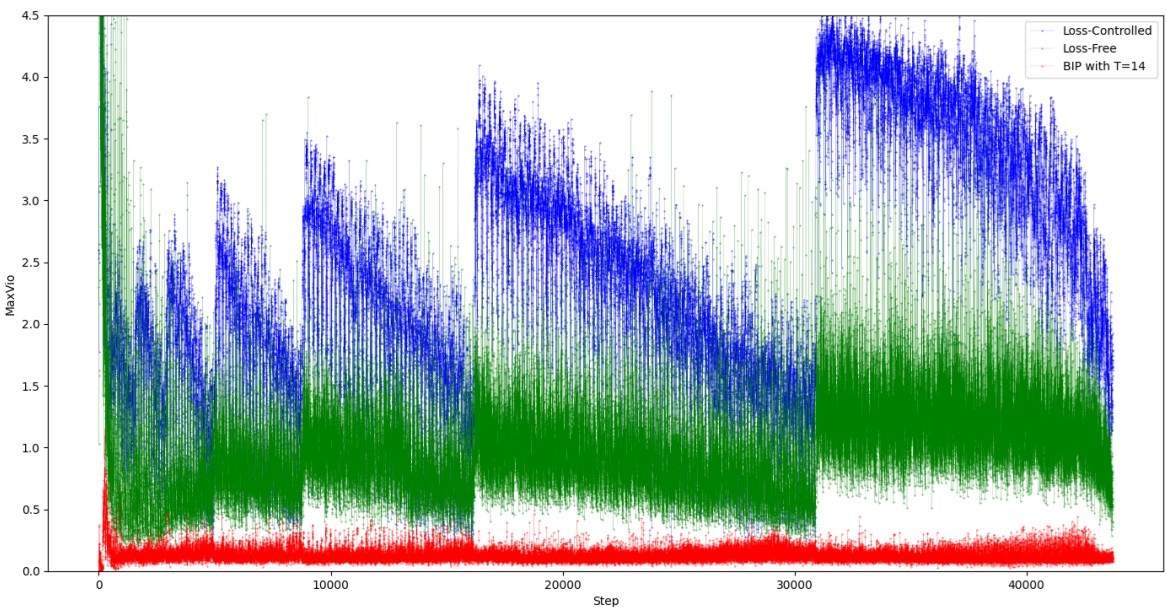

Figure 15: The line graph of relationships between training steps and $MaxVio_{batch_i}$ by different methods in the 64-expert model on layer 5.

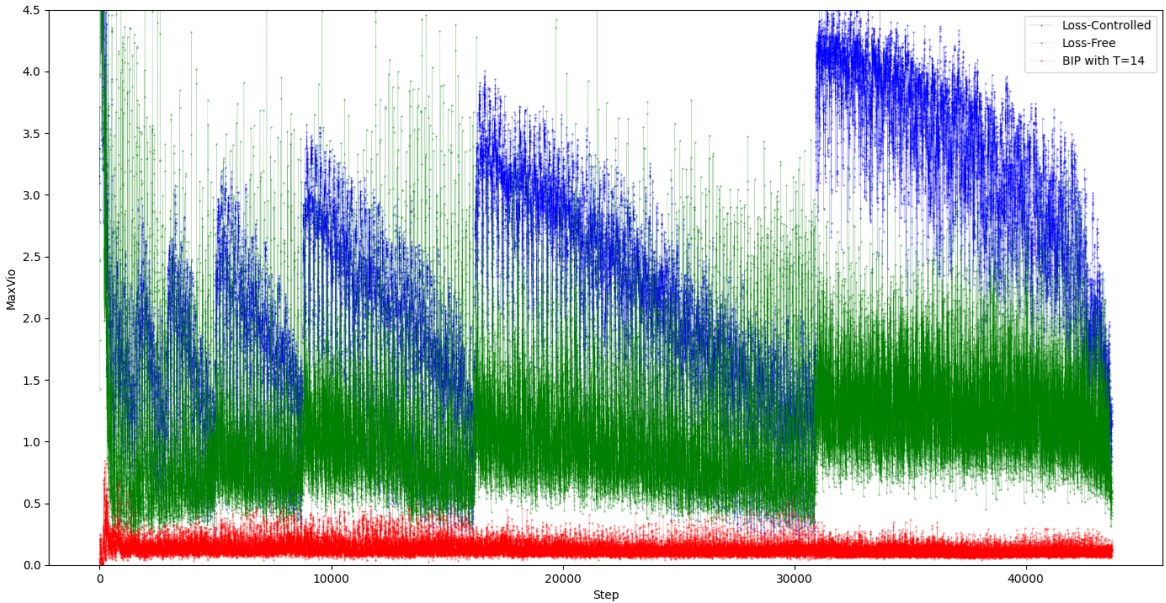

Figure 16: The line graph of relationships between training steps and $MaxVio_{batch_i}$ by different methods in the 64-expert model on layer 6.

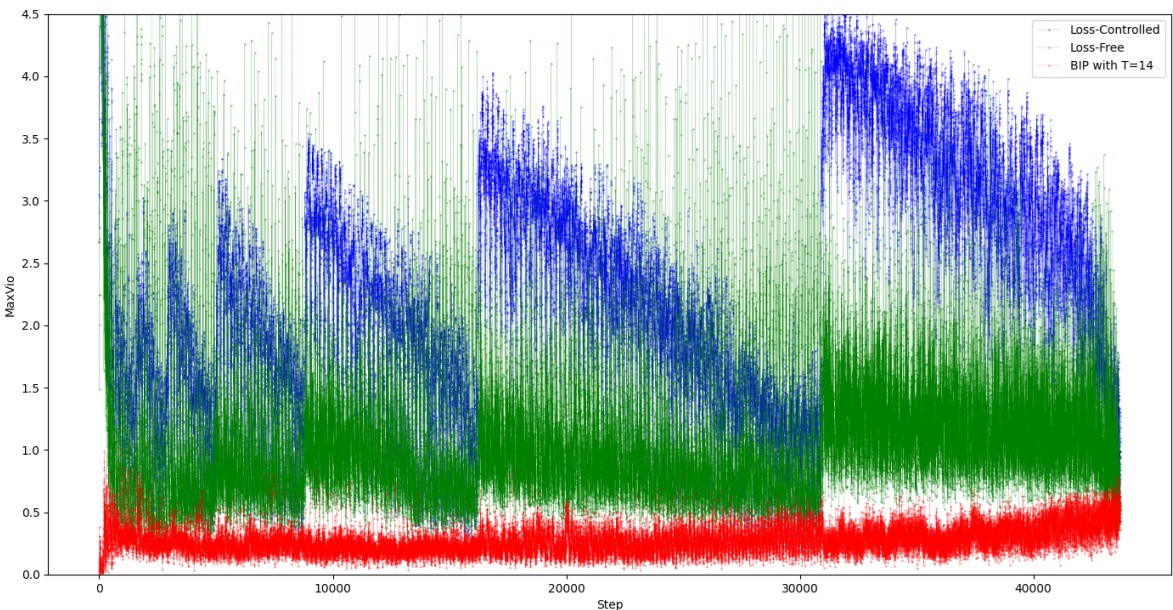

Figure 17: The line graph of relationships between training steps and $MaxVio_{batch_i}$ by different methods in the 64-expert model on layer 7.

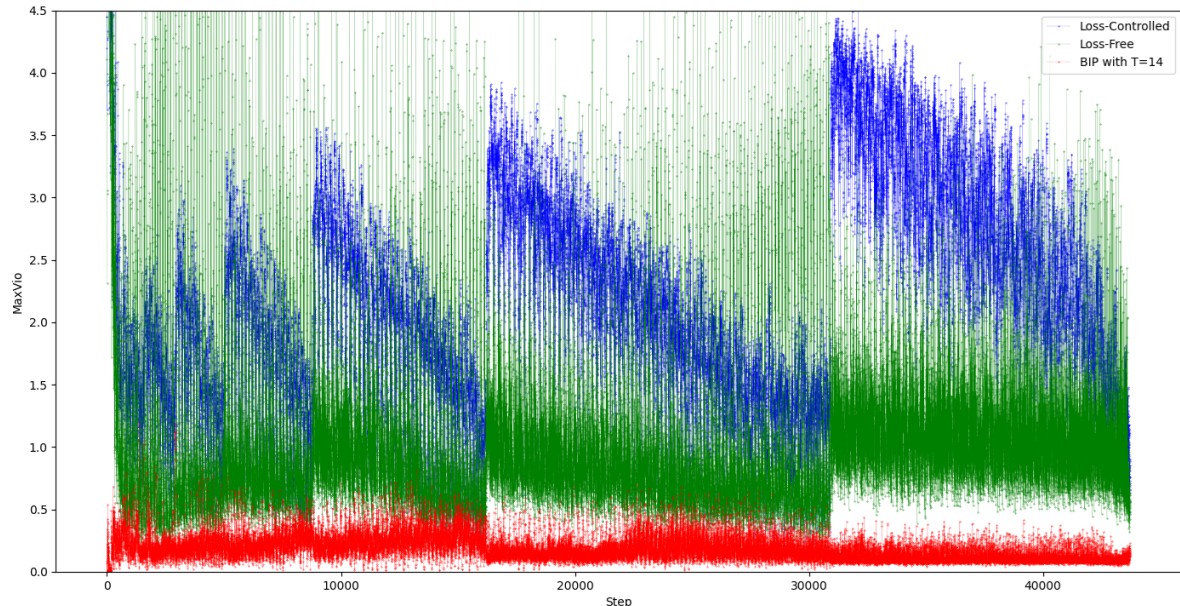

Figure 18: The line graph of relationships between training steps and $MaxVio_{batch_i}$ by different methods in the 64-expert model on layer 8.

