# OpenReview forum: "Binary-Integer-Programming Based Algorithm for Expert Load Balancing in Mixture-of-Experts Models"
_ICLR.cc/2026/Conference — Submitted to ICLR 2026_

### Official Review · Reviewer_t5hG · 2025-10-30

**Soundness:** 2
**Presentation:** 3
**Contribution:** 2
**Rating:** 6
**Confidence:** 3

**Summary:**

This paper proposes an improved version of auxiliary-loss-free load balancing strategy, which is based on binary integer programming. It reported better load balancing control (especially at the beginning) and lower perplexity than auxiliary-loss-controlled ones or the original auxiliary-loss-free method.

**Strengths:**

1. Better load balancing (especially at the beginning).
2. It reported better performance.
3. Algorithm 4 can be applied to cases where large batch sizes and expert parallelism are used.

**Weaknesses:**

1. Can you report the training data amount and the batch size (and the loss curve, if you can update it in the PDF or somewhere)? I cannot find it in the paper, and I’m wondering whether the data and batch size were too small, which might have caused the behavior to be different from normal cases.

2. I wish experiments were conducted on Algorithm 4 (or some similar version that consumes acceptable GPU memory and allows expert parallelism in large-scale training).

3. Will this algorithm cause the bias term *q* to change too quickly, hurting performance and causing training instability?

**Questions:**

1. The template is a little strange, and you might use `\citep` for some cases.

2. Will a 0.1 auxiliary loss alpha be too large for the loss-controlled baseline? (I'm not clear about the implementation, so I'm not sure about the actual effect of this 0.1 value, but you may report the loss and maxvio around 0.1 (like 0.01 / 1.?)).

3. "Especially in the Loss-Controlled method, since there are discussions showing that on the softmax function the Loss-Controlled method works better Su (2025)." Can you provide the original text on the Kexue blog? It isn't on the linked blog.

4. I shall argue that the gating function for the load balancing methods does not have to be the same, as long as all methods report their best results. It should be viewed as an inseparable part of the load balancing methods.

---

### Official Review · Reviewer_5Nwf · 2025-11-01

**Soundness:** 3
**Presentation:** 3
**Contribution:** 2
**Rating:** 2
**Confidence:** 3

**Summary:**

In this paper, the authors propose a novel balancing algorithm called "BIP-Based Balancing." The core idea is to formulate the expert-token assignment as a Binary Integer Program (BIP) designed to maximize the routing scores for the current batch, with a hard constraint on load balance.

Based on this formulation, the authors analyze its Linear Programming (LP) relaxation and corresponding dual problem. By solving this dual problem using the Alternating Direction Method of Multipliers (ADMM) for a small number of iterations (T) at each step, the algorithm derives a set of dual variables. These variables are then used as biases to modify the original routing scores before the Top-K selection.

The authors conduct experiments on two MoE language models (0.3B 16-expert and 1.1B 64-expert) and compare their method against Loss-Controlled (e.g., GShard) and Loss-Free (e.g., DeepSeek-V3) baselines. The results show that BIP-Based Balancing achieves significantly lower load imbalance (AvgMaxVio), better model performance (lower perplexity), and reduced training time.

**Strengths:**

1. Novelty: The formulation of expert balancing as a BIP and the use of dual variables from its LP-relaxation as routing biases is a novel and theoretically sound approach. Personally, I found this formulation very interesting.

2. Good performance: The experimental results look promising. The proposed method outperforms the baselines: it achieves a more stable and balanced load (drastically lower AvgMaxVio/SupMaxVio), results in a better-performing model (lower perplexity), and reduces training time at the same time.

3. Solves the early imbalance issue: A key advantage, clearly shown in Figures 1 and 2, is that this method achieves load balance from the very early training stage. Baselines (blue and green lines) show high initial imbalance or significant fluctuations, while the proposed method (red line) is stable and low from step 0.

**Weaknesses:**

1. CRITICAL: Submission Format Violation: The submitted paper does not appear to follow the standard ICLR template. On this basis alone, the paper is recommended for an **immediate desk rejection**. If AC and other reviewers do not think this is a big issue, I may revise my score.

2. Scalability Concerns: The experiments are conducted on relatively small models (0.3B and 1.1B). While informative, the true test for MoE load balancers is at larger scales and more training steps. Intuitively, if the model is trained for many more tokens and steps, the influence from the early imbalance phase will naturally decrease, as will the relative performance gain of the proposed method.

3. The paper reports a significant (13%) reduction in training time but does not clearly explain why this occurs. The authors could better explicitly state that this algorithmic overhead is negligible compared to the system-level time saved by eliminating "stragglers" (overloaded GPUs that bottleneck parallel training steps). This connection is critical to the paper's main claim.

4. The paper does not state whether the baseline methods (loss-controlled and loss-free) use the token dropping strategy. Token dropping is a common technique to handle imbalance, especially during the early, unstable phases, and should be a good baseline.

5. Missing Training Data Details: The paper does not specify the total size (e.g., number of tokens) of the pre-training dataset, only its source and the max sequence length. This makes it harder to fully assess the reproducibility and trustworthiness of the results.

**Questions:**

1. Could the authors provide a precise measurement of the computational overhead for the proposed routing, lines 7-12? I suspect it is negligible compared to the rest of the training, but it would be better to quantify this.

2. Do the authors have any intuition on why the MaxVio of the loss-controlled and loss-free methods seems to fluctuate periodically during training (as seen in Figures 1 and 2)?

---

### Official Review · Reviewer_gewL · 2025-11-01

**Soundness:** 2
**Presentation:** 3
**Contribution:** 2
**Rating:** 2
**Confidence:** 4

**Summary:**

This paper proposes a new load-balancing strategy for Mixture-of-Experts (MoE) models called BIP-Based Balancing. It formulates expert assignment as a Binary Integer Programming (BIP) problem. The method introduces a per-layer vector q that adjusts the routing scores. This adjustment is achieved by solving a simplified dual optimization problem using ADMM iterations. The approach aims to maintain expert load balance from the start of pre-training. This is in contrast to previous loss-controlled or loss-free methods that converge slowly. Experiments on small-scale MoE variants of the Minimind model (0.3B and 1.1B parameters) show reduced perplexity. They also indicate roughly 13–14\% shorter training time.

**Strengths:**

1. Novel formulation: Modeling MoE load balancing as a BIP optimization offers a fresh theoretical perspective on routing dynamics.
2. Auxiliary-loss-free: The method avoids using extra loss terms. This allows the model to focus on the main objective.

**Weaknesses:**

1. The proposed BIP formulation requires iterative updates at every routing gate. The paper does not provide clear computational complexity or runtime profiling. Thus, it is unclear if this approach is practical for large-scale MoE models with tens or hundreds of billions of parameters.
2. Experiments are conducted only on small MoE models with single-GPU setups (RTX4090/L20). There is no evidence that the algorithm scales to large LLMs or distributed settings. This significantly weakens the empirical claims.
3. The paper presents the BIP problem as solvable via a simple ADMM-like iteration. However, this is mathematically inconsistent. BIP is NP-hard in general. The authors are actually solving the dual of the LP relaxation, not the integer problem itself. There is no proof that their iterative rule recovers a feasible or optimal integer assignment $X_{ij}$. Thus, the method likely produces an approximate heuristic, not an exact BIP solution.
4. The authors invoke ADMM to justify iterative updates for $p, q, r$. However, the augmented Lagrangian $L(p,q,r,u)$ is not explicitly written. No penalty parameter $\lambda$ is defined. There is no convergence guarantee, which normally requires convexity and Lipschitz continuity. The authors assert that lines 2–3 in Algorithm 2 imply lines 7–12 in Algorithm 1. This is not a valid ADMM update derivation; it is a qualitative analogy, not a formal equivalence.

**Questions:**

Please check the weakness

---

### Meta-Review · Area_Chair_mHTX · 2026-01-04

**Summary:**

The paper proposes "BIP-Based Balancing," a routing strategy for Mixture-of-Experts (MoE) models that formulates token-expert assignment as a Binary Integer Programming (BIP) problem. While framed as a BIP solution, the authors solve a dual of the LP relaxation. There is no proof that the iterative updates recover an optimal or even feasible integer assignment. Experiments were restricted to very small models (0.3B and 1.1B parameters) on single-GPU setups. There is no evidence that the method scales to the distributed environments where MoE models are typically deployed.

**Reviewer Concerns:**

There is no rebuttal from the authors. Reviewer gewL notes that the authors claim to solve a BIP (which is NP-hard) but are actually solving an LP relaxation without proving the iterative rule results in a valid integer assignment. There is no evidence the algorithm scales to large-scale distributed settings or massive LLMs. The authors did not compare the method against "token dropping," a standard technique for handling imbalance.

**Reviewer Scores:**

Reviewer gewL provided the most rigorous technical critique, specifically highlighting that the authors claim to solve an NP-hard BIP problem while actually using a dual LP relaxation without proof of optimality or feasibility. Without a rebuttal to clarify the mathematical derivation of the ADMM updates or the augmented Lagrangian, gewL would likely have convinced the other reviewers that the paper’s theoretical foundation is fundamentally flawed. Reviewer t5hG expressed significant uncertainty about whether the small batch sizes and training data led to the observed results. In a discussion, once gewL highlighted the mathematical inconsistencies, t5hG would almost certainly have downgraded their score to align with the consensus on the paper's lack of rigor.

---

### Decision · Program_Chairs · 2026-01-26

Reject